# Comparison of COVID-19 Severity and Mortality Rates in the First Four Epidemic Waves in Hungary in a Single-Center Study with Special Regard to Critically Ill Patients in an Intensive Care Unit

**DOI:** 10.3390/tropicalmed8030153

**Published:** 2023-03-01

**Authors:** Éva Nagy, Péter Golopencza, István Barcs, Endre Ludwig

**Affiliations:** 1Schools of PhD Studies, Semmelweis University, 1085 Budapest, Hungary; 2Bajcsy-Zsilinszky Hospital and Outpatient Clinic, 1106 Budapest, Hungary; 3Department of Internal Medicine and Hematology, Division of Infectology, Semmelweis University, 1088 Budapest, Hungary; 4National Institute of Hematology and Infectious Diseases, South Pest Central Hospital, 1097 Budapest, Hungary

**Keywords:** COVID-19 epidemiology, SARS-CoV-2 variants, COVID-19 waves, COVID-19 critically ill patients, bloodstream infections, ICU

## Abstract

Different variants of coronavirus 2 (SARS-CoV-2), a virus responsible for severe acute respiratory syndrome, caused several epidemic surges in Hungary. The severity of these surges varied due to the different virulences of the variants. In a single-center, retrospective, observational study, we aimed to assess and compare morbidities and mortality rates across the epidemic waves I to IV with special regard to hospitalized, critically ill patients. A significant difference was found between the surges with regard to morbidity (*p* < 0.001) and ICU mortality (*p* = 0.002), while in-hospital mortality rates (*p* = 0.503) did not differ significantly. Patients under invasive ventilation had a higher incidence of bloodstream infection (aOR: 8.91 [4.43–17.95] *p* < 0.001), which significantly increased mortality (OR: 3.32 [2.01–5.48]; *p* < 0.001). Our results suggest that Waves III and IV, caused by the alpha (B.1.1.7) and delta (B.1.617.2) variants, respectively, were more severe in terms of morbidity. The incidence of bloodstream infection was high in critically ill patients. Our results suggest that clinicians should be aware of the risk of bloodstream infection in critically ill ICU patients, especially when invasive ventilation is used.

## 1. Introduction

A new type of coronavirus pandemic causing acute respiratory illness started in China in December 2019 and has rapidly become a global public health emergency of international concern. The new type of coronavirus has been given the name SARS-CoV-2, and the disease it causes is known as coronavirus disease-19 (COVID-19) [1]. According to a World Health Organization (WHO) aggregate report, the disease caused by the virus has affected more than 663 million people worldwide and caused the deaths of more than 6.5 million people as of mid-January 2023 [2]. 

The course of the infection ranges from mild symptoms to fatally severe respiratory failure [3]. The first reports from China described patients with signs of viral pneumonia and fever, coughing, chest discomfort, and dyspnea [4,5,6]. Gastrointestinal symptoms have also been frequently reported [7,8]. In more severe cases, the virus may induce a strong immune response by rapid replication in the alveolar epithelial cells of the lung, resulting in a cytokine storm which in turn causes acute respiratory distress syndrome (ARDS) and respiratory failure [9,10]. Patients over 60 years of age and those with severe comorbidities are at increased risk of developing ARDS [11,12,13] but may also develop multi-organ failure [14,15,16], a leading cause of death in COVID-19 patients [17,18,19]. Secondary bloodstream infections are more common in patients with severe respiratory tract infections [20,21] and may increase the risk of a fatal outcome.

According to a WHO communication, five variants of concern (VOC) of SARS-CoV-2 were identified between the start of the pandemics and December 2021. These are Alpha (B.1.1.7), Beta (B.1.351), Gamma (P.1), Delta (B.1.617.2), and Omicron (B.1.1.529) [22,23]. These strains cause epidemics of varying severity. Such a high mutation frequency is natural for RNA viruses (such as HIV, influenza, and coronaviruses). Lacking a complementary strand to provide for conservatism, the replication errors of the single-stranded nucleic acid of these viruses are not corrected. If this impacts the structure of a viral protein, antigenicity will also be affected, which in turn may lead to the loss of the protection already acquired [24].

Our study aimed at characterizing the epidemiology of the first four SARS-CoV-2 surges in Hungary, as well as identifying the differences between them. We hypothesized that the course of the disease, the need for intensive care unit (ICU) admission, mortality from complications, and symptoms differed between the surges. We also wished to identify factors influencing high ICU mortality rates.

This paper does not include an examination of the effects of co-morbidities and risk factors. The results of our study on the impact of comorbidity and lifestyle risk factors on disease outcome and progression are summarized in another publication [25].

## 2. Materials and Methods

### 2.1. Study Design and Data Sources

Our single-centered, retrospective, observational study was conducted at the Bajcsy-Zsilinszky Hospital (Budapest, Hungary), during the first 4 epidemic waves in Hungary, from 15 March 2020 to 31 December 2021, on patients hospitalized with laboratory-confirmed SARS-CoV-2 infection. The total inpatient capacity of the hospital is 804 beds. The Intensive Care Unit has 18 beds; however, with the rise in the number of COVID-19 patients in the second wave of the pandemic, the ventilation capacity needed to be increased to 38 beds.

The following information was collected from the hospital’s medical IT system (MedScribe, E-Consult 2000 Kft., Debrecen, Hungary): demographic data, co-morbidities, radiology results, days of care, need for ICU admission, methods of ventilation and oxygen supplementation, vaccination status, vaccine types, symptoms, severity, and outcome and, in the case of ICU patients in critical condition, the data from positive blood cultures. Data was collected using Microsoft Excel spreadsheet (Microsoft Corp., Redmond, WA, USA).

To visualize the dynamics of the epidemic, an epidemic curve was constructed, plotting daily changes in the number of cases, indicating the date of positive SARS-CoV-2 tests, and, for infections confirmed before hospital admission, the date of admission. The hospital load curve illustrates the daily evolution of the number of patients requiring ventilation. The start and end dates of each surge were determined on the basis of information from the Epidemiology and Infection Control Department of the National Centre for Public Health (NNK).

Data were collected and analyzed with the permission of the Institutional Committee of Science and Research Ethics and General Ethics.

### 2.2. Definitions

The infection was considered confirmed in patients with nasopharyngeal or lower respiratory tract samples giving positive results in real-time polymerase chain reaction (RT-PCR) or in vitro immunochromatography (rapid antigen test) for the antigen of the virus. The RT-PCR tests were carried out by accredited laboratories. The antigen-based rapid tests were performed and documented by physicians using approved tests.

On admission, after physical examination, patients were classified into five categories based on their nutritional status (1 = underweight; 2 = normal weight; 3 = overweight; 4 = obese, and 5 = severely obese). The relevant category was indicated on the medical record of patients. During data collection, patients were classified into non-obese (categories 1 and 2) and obese (categories 2, 3, and 4) groups based on their nutritional status.

To define the severity of the disease, the criteria set forth in the Therapeutic Manual by the Ministry of Human Resources [26] were used. The Manual establishes severity on a scale of 0 to 4, from asymptomatic to critical. The clinical and pathological diagnosis for the cause of death, as provided by the treating physician and pathologist, was used as the basis for the assessment of COVID-19-related mortality. To identify cases of bloodstream infection (BSI), ventilator-associated pneumonia (VAP), hospital-acquired pneumonia (HAP), urinary tract infection (UTI), and other infections, we used the epidemiological case definitions published in the Official Journal of the European Union [27].

### 2.3. Participants and Study Size

The present study included adult patients over 18 years of age who were diagnosed with an acute infection. Acute illness was defined as cases where the patient was admitted to hospital within 14 days of taking a sample for a coronavirus infection. For patients with multiple episodes of acute infection, the first episode or the one requiring hospitalization was considered. Patients who required hospital admission with post-COVID symptoms following a previous infection were excluded.

After applying the exclusion criteria, 2873 patients requiring at least 24 h of inpatient care were considered during the study period. Hospitalization was mostly necessary in the case of patients with moderate/severe/critical conditions and those whose admission was justified by some underlying condition. In total, 358 critically ill patients were admitted to the ICU for respiratory support. From the COVID-infected patients admitted to our hospital during the study period, 399 were fully immunized, with doses prescribed by the relevant vaccine application instructions.

### 2.4. Blood Culture Sample Collection and Laboratory Procedures

According to the hospital’s protocol, hemocultures were obtained from all critically ill patients within 48 h of ICU admission. Samples were collected in BD Bactec blood culture bottles, with 1 aerobic and 1 anaerobic bottle being used at each sampling occasion. Samples were incubated in a BD-Bactec 9240 (Becton, Dickinson and Company, Sparks, Nevada, USA) automaton. From positive samples, bacteria/fungi were recovered and identified. Resistance was determined and multi-resistant strains identified according to national and international (European Committee on Antimicrobial Susceptibility Testing) recommendations [28,29,30].

### 2.5. Statistical Methods

In this paper, continuous and categorical variables are presented as percentages, with means, medians, standard deviations, and interquartile ranges provided. For the epidemic curves, the numbers of new cases are presented as absolute numbers and 7-day moving averages.

To determine differences between categorical variables, Fisher’s exact test and the χ^2^ test (Pearson’s chi-squared test) were used. Positive probabilities for independent variables affecting ICU mortality were calculated using multivariate logistic regression. The Hosmer–Lemeshow test was applied to assess the goodness of fit of the logistic regression model. In the logistic regression analysis, an odds ratio was calculated to determine the degree of risk, with a 95% confidence interval. The regression model was built using the backward elimination method. After applying the elimination method, the results of the initial and final models were compared. The results of the initial model are reported in this paper wherever the significant variables remaining in the last step of the elimination did not change significantly. Non-significant variables are also provided.

Statistical tests were performed using SPSS Statistics V22 (IBM Corp., Armonk, NY, USA). Two-tailed α values below 0.05 were considered as statistically significant. Our study design and the presentation of results followed the guidelines of Strengthening the Reporting of Observational Studies in Epidemiology (STROBE) [31].

## 3. Results

### 3.1. Epidemiological Description of Epidemic Waves

The epidemiological descriptions of individual surges are summarized in Table 1. The highest number of hospitalized and artificially ventilated infected patients was recorded in Wave III. The median age was the highest in Wave I (75.5 ± 12.67, IQR: 66–84; Table 1). The median number of days of care at the COVID ward exceeded 10 days in all waves, with the number of days of ICU care varying between 6 and 11.

### 3.2. Epidemic Dynamics

The dynamics of an epidemic is illustrated by the absolute numbers and 7-day moving averages of new cases and the 7-day moving averages of mortality cases (Figure 1).

Hospital load trends are illustrated in Figure 2. The peak load was measured on 5 April 2021 at 183 hospitalized patients, of which 24 people were ventilated in the ICU; this number corresponds to 48% of the patients treated in the hospital that day.

### 3.3. Severity of Course and Disease Outcome

The distribution of severity categories in the individual surges showed significant differences. In the majority of cases, severity was moderate, with a shift towards the severe category in the third wave only (Table 2). Rates of mortality due to COVID-19 complications did not show significant differences (*p* = 0.504; Table 2). A detailed summary of symptoms observed during the epidemic waves is presented in Table A1 in Appendix A. The leading symptoms across the whole study period were dyspnea, fever/high temperature, weakness, and cough.

Almost half of the patients admitted to the inpatient COVID-19 ward (1372 patients, 47.8%) required oxygen supplementation during their stay. The highest number and highest proportion of patients requiring oxygen supplementation were observed in Wave III and Wave IV, respectively (545 patients, 48.8% and 359 patients, 50.1%; *p* < 0.001).

### 3.4. Proportions of Vaccinated Patients in the Total Study Population and the Impact of Vaccination on Mortality and ICU Admission

In our study, altogether, 399 fully immunized patients were hospitalized with COVID-19 after the SARS-CoV-2 vaccines had become available during Waves III and IV. Most of them were admitted to the hospital in Wave IV. In Wave III, immunization significantly reduced both mortality and the need for ICU admission (*p* = 0.016 and *p* = 0.033, respectively; Table 3). In Wave IV, immunization had a lesser but still significant effect on the need for ICU admission (*p* = 0.039); however, it did not significantly affect mortality (*p* = 0.8; Table 3).

### 3.5. Distribution of and Mortality among Critically Ill Patients

#### 3.5.1. Need for ICU Admission

In the first two years of the epidemic, the number of ICU patients did not differ significantly compared to previous years; however, mortality showed an upward trend (Figure 3a). In 2020, 16.7% of patients requiring ICU care were confirmed as being infected with SARS-CoV-2; their proportion increased to 37.5% in 2021.

The proportion of patients requiring ICU care compared to the total number of cases is illustrated in Figure 3b. The proportion and number of patients ventilated (14.4%, 161 patients) was the highest in Wave III. The proportions and numbers of ICU patients requiring artificial ventilation in Wave II, Wave IV, and Wave I were 13% (102 patients), 11.6% (83 patients), and 4.7% (12 patients), respectively (*p* < 0.001).

#### 3.5.2. Mortality in Critically Ill Patients

The causes of high ICU mortality were further analyzed. Results concerning possible risk factors are summarized in Table 4. Of the patients studied, 90.2% had some co-morbidities at the time of infection, and 46.6% were obese. Concerning comorbidities, the significant and most common risk factors identified in our previous study (hypertension, diabetes mellitus, cardiovascular disease, cancer, chronic kidney disease) are highlighted. The median age of patients was above 60 years across all surges. A total of 10.1% of patients had received SARS-CoV-2 vaccination before contracting the disease, with most of them having been infected during Wave IV. Ventilator-associated pneumonia (VAP) was confirmed in 73 (28.5%) of the invasively ventilated patients. In addition to these cases, hospital-acquired pneumonia (HAP), urinary tract infection, and other infections (surgical site infections and skin and soft tissue infections in most cases) were confirmed in five, six and fourteen patients across the waves, respectively (Table 4).

A multivariate logistic regression model was used to examine the effect of potential risk factors on ICU mortality (Table 5). The risk effects of BSI and VAP were analyzed separately. Other infections were combined into the category ‘Other infections’ due to the low number of cases. No statistically significant risk factors were identified in Waves I or II. In Wave IV, age had a marginally significant effect on mortality, slightly increasing it (*p* = 0.005). In Wave III, in addition to age, bloodstream infection also became a significant risk factor, causing a profound, more than nine-fold increase in mortality (*p* < 0.001; Table 5). Over the entire study period, BSI patients showed a higher mortality rate compared to the non-BSI group (78.7% vs. 54.1%; *p* < 0.001). Ventilator-associated pneumonia in Wave III was a high yet statistically insignificant risk factor, (*p* = 0.064; Table 5). Other infections were not proven to be significant risk factors regarding the mortality caused by COVID-19 complications.

#### 3.5.3. Bloodstream Infections

In total, bloodstream infections developed in 38% of ICU patients, with the highest rate observed in Wave III (45.3%). Bloodstream infection cases were divided according to cause into monomicrobial and polymicrobial infections (84 and 52 patients, respectively). On average, BSI patients spent more time in the ICU compared to the non-BSI group (12 ± 11.3 vs. 7.6 ± 5.8 days).

The distribution of pathogens and the prevalence of multi-drug resistant (MDR) pathogens are summarized in Table 6. As for the relative frequency of pathogens in hemocultures testing positive, Gram-positive and Gram-negative pathogens as well as fungal strains were identified in 39%, 56.5%, and 4.5% of patients, respectively. Of the pathogens cultured from hemocultures, 58.7% were difficult-to-treat pathogens (methicillin-resistant *Staphylococcus aureus* (MRSA), *Enterococcus faecalis*, *Enterococcus faecium*, *Klebsiella pneumoniae*, *Acinetobacter baumannii*, *Acinetobacter* sp., *Pseudomonas aeruginosa* and *Stenotrophomonas maltophilia*). Of the isolates, 22% contained MDR bacteria (MRSA, vancomycin resistant *Enterococcus* sp. (VRE), extended-spectrum beta-lactamase (ESBL) and AmpC-producing Enterobacterales, and MDR *Acinetobacter* sp.).

#### 3.5.4. Effect of Invasive Ventilation on the Development of Bloodstream Infection

The distribution of ventilation modes used in ICU patients according to epidemic wave is illustrated in Figure 4. Invasive mechanical ventilation (IMV)—divided into the categories IMV, airway pressure release ventilation (APRV), and extracorporeal membrane oxygenation (ECMO) in the figure—was required for 71.5% of patients (256 patients). The proportion of invasively ventilated patients was above 60% across all epidemic waves. A total of seven patients were transferred to another institution for ECMO treatment to make up for their missing lung function (Figure 4).

Table 7 illustrates the proportion of patients with bloodstream infections among invasively ventilated patients. Bloodstream infections occurred more frequently in the IMV group compared to non-IMV patients (*p* < 0.001). A significant difference between the two groups was observed in all surges except for the first one. The effect of invasive ventilation on the development of bloodstream infections was tested by an adjusted regression model. IMV was identified as a significant risk factor when considering all patients (aOR: 8.92 (CI: 4.44–17.95); *p* < 0.001). VAP also significantly increased the odds of developing healthcare-associated bloodstream infection (aOR: 7.95 (4.19–15.09); *p* < 0.001).

## 4. Discussion

In our study, we analyzed the characteristics of the first four waves of the SARS-CoV-2 epidemic in Hungary among hospitalized patients over a long period of time. Our results demonstrate that the surges were significantly different concerning the severity of morbidity, the need for ICU admission and ICU mortality. Furthermore, invasive ventilation and ventilator-associated pneumonia increased the odds of critically ill patients developing bloodstream infections, which in turn significantly increased the overall risk of mortality. There was no statistically significant difference in in-hospital mortality between the surges. As for the symptoms, our results were similar to the findings of other national studies [32,33].

In Hungary, the original variant was responsible for the first two surges. The third wave was caused by the alpha variant and the fourth by the delta variant [34,35]. The first two surges differed in several aspects, probably due to different epidemiological management strategies. Patients in the first wave had the highest mean age and mortality. In the first half of 2020, as a result of strict epidemiological measures, the majority of patients admitted to hospital were of advanced age, had many co-morbidities, and were living in nursing homes, which may explain the high mortality rate observed.

Over the course of the second surge of the epidemic, three times as many patients needed hospitalization and 8.5 times as many ICU patients were ventilated than in the first wave. The average age of patients, the average number of days of care, and the mortality rate were lower in the second wave. Epidemic management was hampered by the high incidence of hospital-acquired outbreaks and the high number of healthcare workers acquiring the disease.

With the emergence of the alpha variant, the third surge started in Week 4 of 2021 [34,35]. The third wave was the most severe in terms of hospital loads and disease progression. The average age of patients dropped below 70 years and nearly 60% were in severe or critical condition. The morbidity rate among health workers was lower, presumably due to the vaccination campaign launched at the beginning of the year and the immunity acquired by contracting the disease in the previous wave.

The fourth surge was dominated by the delta variant. It spread more rapidly than the alpha variant and caused more severe cases, in the unvaccinated population in particular [34]. Hospital loads decreased, probably as a result of the widespread availability of SARS-CoV-2 vaccines. The proportion of critically ill patients requiring ventilation was lower, but mortality among hospitalized and ICU patients remained high. This surge showed the highest number of breakthrough infections. Half of the patients admitted to the hospital had received active immunization before contracting the illness. Our results suggest that in the wave caused by the delta variant, active immunization was no longer a significant protective factor for in-hospital mortality, which prompted the development of further preventive measures and the administration of booster vaccines, especially among the vulnerable and the elderly, as well as health workers.

The disease course of COVID-19 ranges from asymptomatic viral shedding to fatal multi-organ dysfunction. The related risk factors and genetic predisposing factors influencing the progression of the disease are the subject of much research. Turk et al. divided disease progression into three clinico-biological phases: 1. initial phase also known as the asymptomatic or presymptomatic phase; 2. propagation phase, with mild/moderate/severe respiratory symptoms; 3. complication phase, a multisystemic clinical syndrome with impaired and/or defective immunity. The third clinical phase, manifested in multi-organ failure, septic shock, and ARDS, may be considered as COVID-19 syndrome due to the complex clinical course. This study demonstrated that the different phases have different genomic features, which in turn lead to different clinical symptoms driven by different mechanisms [36].

In comparison with our previous study, some differences were observed in the mortality risk factors identified in the general population and ICU patients. In addition to age and male sex, obesity, and the presence of certain comorbidities (e.g., cardiovascular disease, cancer, chronic renal failure) were significant risk factors for all patients included in the study [25]. Of the ICU patients studied, 90% had comorbidities and nearly half of them were obese, yet these factors did not impair survival to a statistically significant extent. Of the 2873 patients, 12.5% were admitted to the ICU due to the severity of their condition and/or the need for ventilator support. Among the 71.5% of ICU patients who required invasive mechanical ventilation, the mortality rate was 80.5%. The risk effect of invasive ventilation on mortality could be statistically demonstrated. Besides invasive ventilation, age and bloodstream infection were significant risk factors. In our study, the incidence of bloodstream infection in critically ill patients was high; this finding is supported by similar results reported by other studies [20,21,37,38].

High ICU mortality may be explained by several factors. Much like in many other hospitals in Hungary, the number of days patients spent in ICU increased significantly during the COVID epidemic [39]. Combined with the increased use of invasive techniques and medical devices typical of COVID care, long hospital stay significantly increases the risk of infection. Across all surges, the need for invasive ventilation was high, and this method is a known risk factor for bacterial superinfection of the lung and subsequent bloodstream infection. In our study, we identified invasive ventilation as a significant risk factor for healthcare-associated bloodstream infections. A proportion of bloodstream infections (14.5%) developed prior to invasive ventilation, suggesting that primary bloodstream infections may have increased the severity of illness, making IMV use in these cases forced and secondary. However, given that in the majority of cases BSI developed after invasive ventilation, we consider that the statistically demonstrated risk role of IMV is valid. Furthermore, the immune dysfunction induced by severe SARS-CoV-2 infection and the immunosuppressive effect of prolonged steroid treatment as part of COVID-19 therapy may predispose patients to concurrent infections [10,40,41]. ICU load was high, especially in the third wave. Increased workload meant healthcare workers were forced to deal with more patients daily, which led to a decrease in their infection control compliance. This in turn may have contributed to the increased incidence of BSI. A recent publication has shown that SARS-CoV-2 causes dysbiosis of the gut microbiome, resulting in the increased translocation of gut bacteria into the bloodstream, causing severe, potentially mortal secondary sepsis [42]. This may explain our observation of a high proportion of bloodstream infections in ICU patients having been caused by the gut microbiota.

There are some limitations to this study. Being a single-center study, it cannot be generalized to the entire Hungarian population nor to the characteristics of COVID-19 care in Hungary. In addition to IMV and VAP, the use of invasive devices (e.g., central venous catheters and peripheral vascular catheters) may have increased the risk of bloodstream infections, but the present study did not include an assessment of device use rates and their risk impact due to limited data. The statistical analysis of the data from the first surge is of limited value due to the low number of cases, especially regarding ICU mortality. The study did not include a follow-up of therapy and laboratory parameters; thus, the extent to which long-term steroid treatment might have influenced the risk of bloodstream infections cannot be assessed.

## 5. Conclusions

The excess infections observed across all surges manifested in the form of severe secondary infections in the ICU, often causing multi-organ failure. Invasively ventilated SARS-CoV-2 patients have a higher risk of developing bloodstream infections, which in turn increases mortality risks. Therefore, the close monitoring of patients and increased adherence to hygiene protocols may improve patient survival.

## Figures and Tables

**Figure 1 tropicalmed-08-00153-f001:**
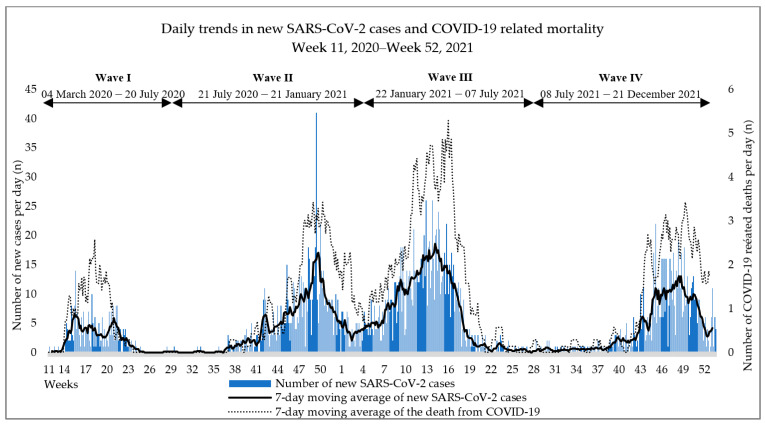
Epidemic dynamics.

**Figure 2 tropicalmed-08-00153-f002:**
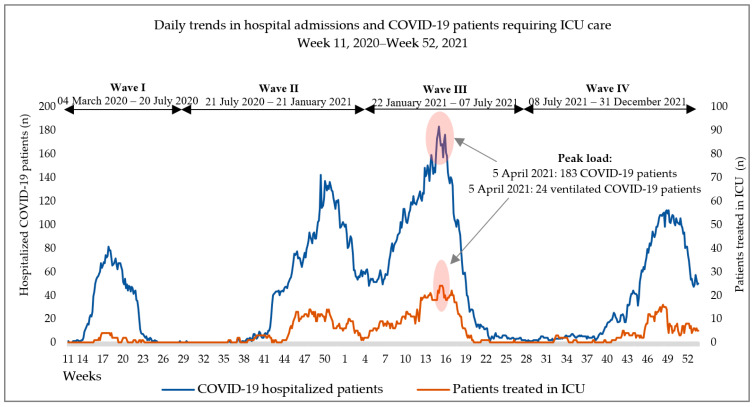
Hospital loads during the epidemic waves I–IV.

**Figure 3 tropicalmed-08-00153-f003:**
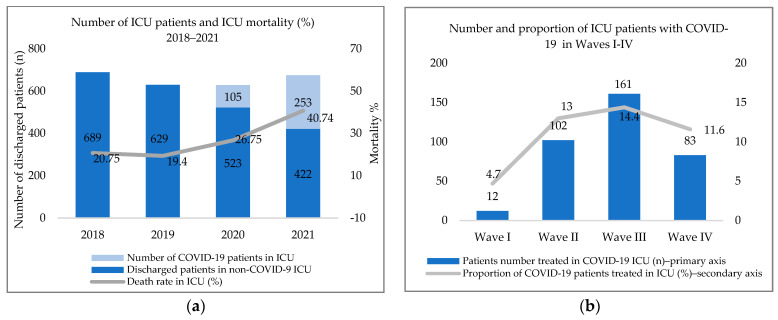
(**a**) Number of and mortality among ICU patients between 2018 and 2021. (**b**) Number and proportion of ICU patients with COVID-19 in Waves I–IV.

**Figure 4 tropicalmed-08-00153-f004:**
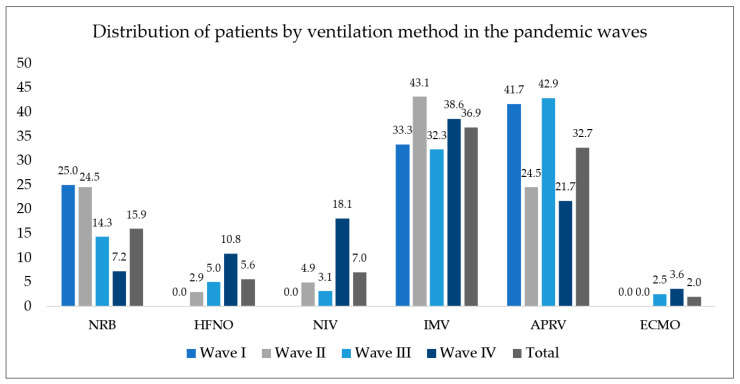
Distribution of ICU patients treated by ventilation method in the epidemic waves. NRB—non-rebreather mask; HFNO—high-flow nasal oxygen; NIV—non-invasive ventilation; IMV—invasive mechanical ventilation; APRV—airway pressure release ventilation; ECMO—extracorporeal membrane oxygenation.

**Table 1 tropicalmed-08-00153-t001:** Epidemiological description of epidemic waves.

	Wave I ^a^	Wave II ^b^	Wave III ^c^	Wave IV ^d^	Total
Number of SARS-CoV-2 cases (n/%)	256 (8.9%)	785 (27.3%)	1116 (38.8%)	716 (24.9%)	2873
Median age ± SD (IQR, min–max)	75.5 ± 12.67 (66–84)	71 ± 15.14 (61.5–80)	67 ± 15.64 (54–76)	69 ± 15.94 (57–79)	69 ± 15.57 (58–79)
Male (n/%)	115 (44.9%)	391 (49.8%)	564 (50.5%)	355 (49.6%)	1425 (49.6%)
Mean number of days of care at the COVID-19 ward ± SD (IQR, min–max)	15.27 ± 11.89 (5–24)	12.1 ± 11.61 (3–17)	11.15 ± 6.97 (7–14)	10.08 ± 6.91 (6–13)	11.51 ± 9.06 (6–15)
Number of ICU patients (n/%)	12 (4.7%)	102 (13%)	161 (14.4%)	83 (11.6%)	358 (12.5%)
Mean number of days of ICU care ± SD (IQR, min–max)	6.58 ± 5.50 (2.25–10.25)	11.57 ± 11.50 (3–15.25)	9.96 ± 7.42 (4–13)	10.27 ± 8.87 (4–14)	10.38 ± 9.06 (4–13.25)

SD—standard deviation; IQR—interquartile range; ICU—intensive care unit. Data for continuous and categorical variables are presented as means or medians ± SD and as percentages, respectively. Starting and end dates of waves: ^a^ 15 March 2020–20 July 2020 (128 days); ^b^ 21 July 2020–21 January 2021 (185 days); ^c^ 22 January 2021–7 July 2021 (167 days); ^d^ 8 July 2021–31 December 2021 (177 days).

**Table 2 tropicalmed-08-00153-t002:** Distribution of disease severity and outcome in the epidemic waves I–IV given as number and percentage of cases.

Severity	Wave I	Wave II	Wave III	Wave IV	*p* Value
Asymptomatic	37 (14.5%)	138 (17.6%)	116 (10.4%)	49 (6.8%)	<0.001
Mild	24 (9.4%)	78 (9.9%)	59 (5.3%)	89 (12.4%)	<0.001
Moderate	170 (66.4%)	354 (45.1%)	274 (24.6%)	266 (37.2%)	<0.001
Severe	13 (5.1%)	128 (16.3%)	521 (46.7%)	234 (32.7%)	<0.001
Critical	12 (4.7%)	87 (11.1%)	146 (13.1%)	78 (10.9%)	0.002
Radiologically confirmed pneumonia	146 (57%)	476 (60,6%)	850 (76.2%)	468 (65.4%)	<0.001
Outcome					
Recovered	70 (27.3%)	336 (42.8%)	656 (58.8%)	412 (57.5%)	<0.001
Died	75 (29.3%)	196 (25%)	295 (26.4%)	197 (27.5%)	0.504
Hospital discharged—not recovered	111 (43.4%)	253 (32.2%)	165 (14.8%)	107 (14.9%)	<0.001

**Table 3 tropicalmed-08-00153-t003:** Impact of vaccination on mortality due to complications and on the need for ICU admission in Waves III and IV.

	Vaccinated (n/%)	aOR [95% CI]	
		Mortality	ICU Admission
Wave III.	42 (3.8%)	0.39 * [0.18–0.84]	0.11 * [0.02–0.84]
Wave IV.	357 (49.9%)	0.94 [0.56–1.56]	0.69 * [0.48–0.98]

aOR—adjusted odds ratio; CI—confidence interval; The correlation was calculated by logistic regression, adjusted to age, sex, comorbidities, and obesity; results were significant at *p* < 0.05. Values in square brackets indicate the 95% confidence interval in each correlation. * *p* < 0.05.

**Table 4 tropicalmed-08-00153-t004:** Demographics of ICU patients, mortality and prevalence of factors potentially influencing mortality.

	Wave I	Wave II	Wave III	Wave IV	Total
Number of COVID-19 cases	12	102	161	83	358
Median age ± SD (IQR, min–max)	68 ± 9.57 (58.75–73.75)	68 ± 10.99 (62–75)	67 ± 12.57 (58–72)	62 ± 14.15 (51–71)	66 ± 12.65 (58–73)
Male (n/%)	8 (66.7%)	71 (69.6%)	95 (59%)	46 (55.4%)	220 (61.5%)
Mortality (n/%)	5 (41.7%)	52 (51%)	117 (72.7%)	53 (63.9%)	227 (63.4%)
Vaccination (n/%)	0	0	1 (0.6%)	35 (42.2%)	36 (10.1%)
Presence of Comorbidity (n/%)	12 (100%)	95 (93.1%)	148 (91.9%)	68 (81.9%)	323 (90.2%)
Obesity (n/%)	6 (50%)	41 (40.2)	79 (49.1%)	41 (49.4%)	167 (46.6%)
Hypertonia (n/%)	8 (66.7%)	75 (73.5%)	112 (69.6%)	49 (59%)	244 (68.2%)
Diabetes mellitus (n/%)	6 (50%)	40 (39.2%)	49 (30.4%)	28 (33.7%)	123 (34.4%)
Cardiovascular disease (n/%)	8 (66.7%)	46 (45.1%)	59 (26.6%)	25 (30.1%)	138 (38.5%)
Cancer (n/%)	2 (16.7%)	10 (9.8%)	25 (15.5%)	7 (8.4%)	44 (12.3%)
Chronic kidney disease (n/%)	3 (25%)	13 (12.7%)	11 (6.8%)	4 (4.8%)	31 (8.7%)
BSI (n/%)	1 (8.3%)	34 (33.3%)	73 (45.3)	28 (33.7)	136 (38.0%)
VAP (n/%) *	0	20 (29)	33 (26.4)	20 (37.7)	73 (28.5)
HAP (n/%)	0	1 (1.0)	2 (1.2)	2 (2.4)	5 (1.4)
UTI (n/%)	0	4 (3.9)	1 (0.6)	2 (2.4)	6 (1.7)
Other infections ^#^ (n/%)	0	5 (4.9)	8 (5.0)	1 (1.2)	14 (3.9)

SD—standard deviation; IQR—interquartile range; BSI—bloodstream infection; VAP—ventilator-associated pneumonia; HAP—hospital-acquired pneumonia; UTI—urinary tract infection. * expressed as the percentage of invasively ventilated patients; ^#^ surgical site infections, skin and soft tissue infections, abscess, parotitis, peritonitis, empyema, intra-abdominal infection, Fournier’s gangrene, etc.

**Table 5 tropicalmed-08-00153-t005:** Association between age, sex, vaccination, obesity, the presence of comorbidities, BSI, VAP, and other infections and death due to COVID-19 complications in ICU over the four epidemic waves.

	aOR [95% CI]				
	Wave I	Wave II	Wave III	Wave IV	Total
Age	1.04 [0.90–1.19]	1.01 [0.96–1.05]	1.04 * [1.00–1.08]	1.07 * [1.02–1.13]	1.03 * [1.01–1.05]
Male sex	0.53 [0.015–9.92]	2.54 [0.98–6.58]	0.93 [0.41–2.05]	1.75 [0.55–5.59]	1.28 [0.79–2.06]
Vaccination	N.A.	N.A.	N.A.	0.57 [0.20–1.66]	0.81 [0.39–1.67]
Comorbidity present	N.A.	3.73 [0.61–22.71]	1.88 [0.44–8.15]	1.48 [0.38–5.72]	1.59 [0.74–3.44]
Obesity	2.35 [0.16–34.19]	1.26 [0.52–3.04]	1.49 [0.64–3.49]	1.63 [0.55–4.86]	1.54 [0.95–2.50]
Hypertonia	N.A.	0.49 [0.18–1.36]	1.09 [0.45–2.60]	0.55 [0.17–1.77]	0.75 [0.44–1.26]
Diabetes mellitus	N.A.	1.02 [0.42–2.43]	1.05 [0.45–2.45]	1.34 [0.42–4.36]	1.02 [0.62–1.67]
Cardiovascular disease	N.A.	1.58 [0.64–3.91]	0.98 [0.41–2.38]	1.14 [0.29–4.49]	1.20 [0.72–2.00]
Cancer	N.A.	2.54 [0.57–11.40]	1.41 [0.48–4.18]	1.34 [0.18–10.20]	1.73 [0.83–3.60]
Chronic kidney disease	N.A.	2.75 [0.71–10.70]	0.79 [0.19–3-49]	0.08 [0.00–1.56]	0.85 [0.38–1.91]
BSI	N.A.	1.90 [0.79–4.53]	9.72 ** [3.68–25.67]	1.06 [0.37–3.05]	3.32 ** [2.01–5.48]
VAP	N.A.	1.98 [0.55–7.11]	7.38 [0.89–64.41]	0.75 [0.12–4.77]	1.86 [0.87–3.98]
Other infections ^#^	N.A.	0.61 [0.16–2.36]	1.22 [0.31–4.91]	1.65 [0.26–10.64]	0.89 [0.41–1.94]

aOR—adjusted odds ratio; CI—confidence interval; N.A.—not applicable; BSI—bloodstream infection; VAP—ventilator-associated pneumonia. The correlation was calculated by logistic regression; results were significant at *p* < 0.05. Values in square brackets indicate the 95% confidence interval in each correlation. * *p* < 0.05 and ** *p* < 0.001. ^#^ other infections: urinary tract infection, hospital-acquired pneumonia, surgical site infection, skin and soft tissue infection, intraabdominal infection, etc.

**Table 6 tropicalmed-08-00153-t006:** Distribution of pathogens in ICU patients with bloodstream infections (n = 136) and prevalence of bacteria identified as contaminants.

Identified Microorganisms in Bloodstream Infections (N = 223)	Number of Microorganisms (N/%)	MDR (N/%)
*Staphylococcus aureus*	26 (11.7%)	12 (46.2%)
Coagulase-negative staphylococci (CoNS) *	7 (3.1%)	-
*Streptococcus pneumoniae*	2 (0.9%)	-
Other Streptococcus sp.	5 (2.2%)	-
*Enterococcus faecalis*	27 (12.1%)	-
*Enterococcus faecium*	16 (7.2)	6 (37.5%)
*Escherichia coli*	6 (2.7%)	2 (33.3%)
*Klebsiella pneumoniae*	22 (9.9%)	13 (59.1%)
*Klebsiella aerogenes*	7 (3.1%)	3 (42.9%)
*Enterobacter cloacae*	13 (5.8%)	5 (38.5%)
Other Enterobacter sp.	9 (4.0%)	3 (33.3%)
*Citrobacter* sp.	2 (0.9%)	-
*Proteus* sp.	4 (1.8%)	-
*Acinetobacter baumannii*	11 (4.9%)	4 (36.4%)
Other Acinetobacter sp.	5 (2.2%)	1 (20%)
*Pseudomonas aeruginosa*	18 (8.1%)	-
*Serratia marcescens*	6 (2.7%)	-
*Stenotrophomonas maltophilia*	20 (9.0%)	-
Other Gram-positive	4 (1.8%)	-
Other Gram-negative	3 (1.3%)	-
*Candida albicans*	5 (2.2%)	-
Other Candida sp.	5 (2.2%)	-
Bacteria identified as contaminants (n = 171)		
Coagulase-negative staphylococci (CoNS)	157 (91.8%)	-
*Corynebacterium* sp.	6 (3.5%)	-
*Micrococcus* sp.	3 (1.8%)	-
*Peptococcus* sp.	1 (0.6%)	-
*Cutibacterium* sp. (*Propionibacterium* sp.)	3 (1.8%)	-
Gram-positive rods	1 (0.6%)	-

MDR—multi-drug resistant. * In the case of coagulase-negative staphylococci, a case was considered as a laboratory-confirmed bloodstream infection if the same coagulase-negative staphylococci could be recovered from blood culture samples taken on two or more separate occasions, but within a maximum of 48 h; the patient also had at least one of the symptoms including fever (>38°), chills, or hypotension and these symptoms could not be explained by any other cause.

**Table 7 tropicalmed-08-00153-t007:** Proportion of IMV patients in the BSI and non-BSI groups.

	Invasive Mechanical Ventilation (IMV)		*p* Value ^†^
	BSI Group *	Non-BSI Group **	
Wave I	1/1 (100%)	8/11 (72.7%)	1.0
Wave II	29/34 (85.3%)	40/68 (58.8%)	0.007
Wave III	72/73 (98.6%)	53/88 (60.2%)	<0.001
Wave IV	24/28 (85.7%)	29/55 (52.7%)	0.003
Total	126/136 (92.6%)	130/222 (58.6%)	<0.001

BSI—bloodstream infection; ^†^ Chi square test and Fisher’s Exact test as appropriate. * (*n*_BSI-IMV_/*n*_total-BSI_ %) ** (*n*_nonBSI-IMV_/*n*_total-non-BSI_ %).

## Data Availability

The data sets used and/or analyzed in the current study are available upon reasonable request to the corresponding author.

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
