# Peer review of "Comparison of COVID-19 Severity and Mortality Rates in the First Four Epidemic Waves in Hungary in a Single-Center Study with Special Regard to Critically Ill Patients in an Intensive Care Unit"

_tropicalmed, 2023, doi:10.3390/tropicalmed8030153_

Round 1

Reviewer 1 Report

Peer-review on manuscript:

“Comparison of COVID-19 Severity and Mortality Rates in the First Four Epidemic Waves in Hungary in a Single-Center Study with Special Regard to Critically Ill Patients in an Intensive Care Unit

The reviewed paper describes single-center, retrospective, observational study of the first four waves of the SARS-CoV-2 epidemic in Hungary among hospitalized patients. Morbidities and mortality rates in the epidemic waves I to IV in critically ill patients was compared.  Risk of bloodstream infections and its correlation with invasive ventilation  discussed. The paper contains a lot of data that is interesting to the reader, but statistical analysis is sometimes questionable.

Reviewer 2 Report

The manuscript describes important novel data in the studied area and worth to be published with some improvements in the presentation style and language. Discussion section shall be expanded via additional perspectives regarding previous papers in the COVID-19 field particularly focusing on the 3 clinicobiological determination of the disease severity (such as; 

Three critical clinicobiological phases of the human SARS-associated coronavirus infections.

by Turk C, and coworkersEur Rev Med Pharmacol Sci. 2020 Aug;24(16):8606-8620. doi: 10.26355/eurrev_202008_22660.PMID: 32894568 

Reviewer 3 Report

Dear editor,

I have reviewed the manuscript “Comparison of COVID-19 Severity and Mortality Rates in the First Four Epidemic Waves in Hungary in a Single-Center Study with Special Regard to Critically Ill Patients in an Intensive Care Unit" submitted to “Trop. Med. Infect. Dis” for publication. In this paper, authors aim to described the invasive ventilation in COVID-19 critically ill patients hospitalized in the ICU and the role of invasive ventilation in increasing patients' hospital infections and its relationship with increased mortality.

Comment to the authors,

Line 36:  Please write the abbreviation of the World Health Organization. World Health Organization (WHO)…also for MDR (Line 224,230-231)

Line 67: How many beds does the hospital's ICU have? Please mention in the text.

Line 90-91: The word rt-PCR should be replaced by RT-PCT.

Table 3 and 4: For easy access of the readers, it is suggested to add the important risk factors in the previous study to one of the tables.

Was tocilizumab prescribed to patients hospitalized in the ICU and at what waves? As you know, the administration of this drug increases the possibility of secondary bacterial infection in patients, and it is important to mention this in this study.

Authors should show the rates of all types of ICU infections in 4 waves in Table 3 or 4. For example, UTI 20% HAP 10% OR VAP 30% ,….

Considering the type of study, it is necessary to investigate HAP and VAP in patients.

Table 5: Please check the MRSA again for MDR.

Why only the IMV factor has been investigated to be related to the patient's blood infection? If possible, it is suggested to compare the use of invasive tools such as CVC line etc. for ICU patients in every 4 waves because the rate of blood infection is reported to be high and whether the use of these tools is a risk factor for infection.

Round 2

Reviewer 3 Report

Dear editor,

Thank you and dear authors. Corrections and suggestions are well done. Only the abbreviations should be checked once again in the full text.

Comment to the authors,

Line 265: multi-drug resistant Acinetobacter should be replaced with MDR Acinetobacter.

Line 276: invasive mechanical ventilation (IMV), should be replaced with IMV.

Line 292: Ventilator-associated pneumonia should be replaced with VAP.